# The impact of social and environmental extremes on cholera time varying reproduction number in Nigeria

Gina E. C. Charnley [1,2]*, Sebastian Yennan[3], Chinwe Ochu[3], Ilan Kelman[4,5,6], Katy A. M. Gaythorpe[1,2], Kris A. Murray[1,2,7]

**1** Department of Infectious Disease Epidemiology, School of Public Health, Imperial College London, London, United Kingdom, **2** MRC Centre for Global Infectious Disease Analysis, School of Public Health, Imperial College London, London, United Kingdom, **3** Surveillance and Epidemiology Department/IM Cholera, Nigeria Centre for Disease Control, Abuja, Nigeria, **4** Institute for Risk and Disaster Reduction, University College London, London, United Kingdom, **5** Institute for Global Health, University College London, London, United Kingdom, **6** University of Agder, Kristiansand, Norway, **7** MRC Unit The Gambia at London School of Hygiene and Tropical Medicine, Fajara, The Gamiba

* g.charnley19@imperial.ac.uk

**Data Availability Statement:** All data used here are taken from public sources and referenced throughout, except the cholera data which was provided by the NCDC and contains sensitive

## Abstract

Nigeria currently reports the second highest number of cholera cases in Africa, with numerous socioeconomic and environmental risk factors. Less investigated are the role of extreme events, despite recent work showing their potential importance. To address this gap, we used a machine learning approach to understand the risks and thresholds for cholera outbreaks and extreme events, taking into consideration pre-existing vulnerabilities. We estimated time varying reproductive number (R) from cholera incidence in Nigeria and used a machine learning approach to evaluate its association with extreme events (conflict, flood, drought) and pre-existing vulnerabilities (poverty, sanitation, healthcare). We then created a traffic-light system for cholera outbreak risk, using three hypothetical traffic-light scenarios (Red, Amber and Green) and used this to predict R. The system highlighted potential extreme events and socioeconomic thresholds for outbreaks to occur. We found that reducing poverty and increasing access to sanitation lessened vulnerability to increased cholera risk caused by extreme events (monthly conflicts and the Palmers Drought Severity Index). The main limitation is the underreporting of cholera globally and the potential number of cholera cases missed in the data used here. Increasing access to sanitation and decreasing poverty reduced the impact of extreme events in terms of cholera outbreak risk. The results here therefore add further evidence of the need for sustainable development for disaster prevention and mitigation and to improve health and quality of life.

## Introduction

Cholera was reintroduced into Africa in the 1970s during the seventh and continuing cholera pandemic. It has since caused significant mortality and morbidity, especially amongst the most vulnerable, such as children under five [1]. Despite this, other disease outbreaks have drawn

information and is therefore not available without signing a data sharing agreement with NCDC. The data can be requested via the Nigeria Centre for Disease Control and the data here were obtained from the Surveillance and Epidemiology Department/IM Cholera. The email for NCDC is: info@ncdc.gov.ng The institutional website is: https://ncdc.gov.ng.

**Funding:** This work was supported by the Natural Environmental Research Council [NE/S007415/1] (https://www.ukri.org/councils/nerc/), and the grant was awarded to G.Charnley as part of the Science and Solutions for a Changing Planet DTP. G.Charnley also acknowledges funding from the MRC Centre for Global Infectious Disease Analysis [MR/R015600/1], jointly funded by the UK Medical Research Council (MRC) and the UK Foreign, Commonwealth & Development Office (FCDO), under the MRC/FCDO Concordat agreement and is also part of the EDCTP2 programme supported by the European Union. The funders had no role in study design, data collection and analysis, decision to publish, or preparation of the manuscript.

**Competing interests:** The authors have declared that no competing interests exist.

attention away from cholera in Africa in recent years, including COVID-19 and Ebola [2, 3]. Explosive cholera outbreaks are not uncommon due to the short incubation period (2 hours to 5 days) and high numbers of asymptomatic infections, which when contaminating the environment can sustain transmission [4]. Cholera is considered a disease of inequity and is preventable through wide-spread access to safe drinking water and sanitation [5]. However, the effect of these pre-existing vulnerabilities on disease risk can be exacerbated in times of environmental and social extremes, which can in turn act as a catalyst for, or exacerbate the impacts of, outbreaks.

Previous research has found several links between extreme events and cholera including floods, drought and conflict [6–8]. Disaster-related risk factors leading to disease outbreaks include an inability to access routine care such as vaccination, fears over safety, destruction of infrastructure, disruption of water, sanitation and hygiene (WASH) services and human displacement [9, 10]. Environmental risk factors also act directly on the pathogen and its behaviour, including pathogen dispersal, elevated concentrations due to high temperatures and low precipitation and sustained environmental reservoirs due to the presence of crustaceans [7, 11]. Previous research on disaster-related infectious disease outbreaks have examined extreme events in isolation [7, 10], while others do not include multiple pre-existing socio-economic factors into the methodology [12, 13]. Research linking several social and environmental extremes to diseases and further understanding the complex array of risk factors involved, is a global research gap and is important for predicting cholera transmission and mitigating outbreaks [14].

Nigeria currently reports the second highest number of estimated cholera cases in Africa [1, 15] and has experienced many large outbreaks [16–19]. The high burden is likely due to the presence of many underlying social and environmental risk factors, including a favourable climate [20, 21], poor access to WASH [22, 23] and a high proportion of the population living in poverty (62% at <$1.25/day) [24–26]. It also has a relatively robust reporting system which may correlate with more cases, as cholera is an under-reported disease and cases and deaths are often missed or misattributed. The country has been frequently challenged by both social and environmental extremes such as drought and floods, which may alter in intensity and frequency with climate change [14, 25], along with ongoing conflict in the northeastern region due to Boko Haram (Islamic State West Africa Province) [8, 14]. Due to the ongoing presence of these extremes in Nigeria (conflict and environmental change), it is important to understand their specific effects in terms of health, to protect the population and inform policy.

Here, we aim to expand the current understanding of the role of extreme events in causing or contributing to cholera and increase the attention on cholera in Nigeria. In collaboration with the Nigeria Centre for Disease Control (NCDC), we evaluated by way of machine learning how a range of environmental and social covariates influence cholera through time-varying reproductive number (R). We take advantage of the predictive capacity of machine learning techniques and use R in a novel application to understand the complexities of disaster-related risk factors on cholera outbreak evolution, rather than case and deaths numbers. The originality of the data used here are important, as modelling and testing cholera assumptions across multiple data sources are important to improve our understanding of cholera dynamics. Using the model with the best predictive power, we nowcasted a traffic-light system of cholera risk to illustrate how disasters and pre-existing vulnerabilities alter R in Nigeria, stating specific quantitative thresholds and triggers. Cholera predictions using hypothetical scenarios are a global research gap, and we make use of our novel approach to fill this gap. We anticipate that this relatively simple framework of cholera outbreak risks could be employed across research in fragile settings to understand region, disaster and disease specific risk factors and outbreak triggers.

## Materials & methods

### Ethics statement

The datasets and methods used here were approved by Imperial College Research Ethics Committee and a data sharing agreement between NCDC and the authors. Formal consent was not obtained for individuals in the data used here, as the data were anonymised.

### Datasets

Cholera data were obtained from NCDC and contained surveillance linelist data for 2018 and 2019. The data were age and sex-disaggregated, on a daily temporal scale and to administrative level 4. The data also provided information on the outcome of infection and whether the patient was hospitalised. The data were subset to only include cases that were confirmed either by rapid diagnostic tests or by laboratory culture and only these confirmed cases were used in the analyses. To test if removing suspected cases bias the results and to prove model robustness, a sensitivity analysis was completed running the analysis on all the cholera data (confirmed and suspected), further details and the results are shown in S1 Text. Additionally, NCDC provided oral cholera vaccination (OCV) data. The data were represented by the campaign start and end date, the location (administrative level 1) and the coverage. OCV was transformed to an annual binary outcome variable (0–1) for each state (e.g., if coverage was 100% in a specific year and state, the data point was assigned 1).

A range of covariates were investigated based on previously understood cholera risk factors. Covariates included factors related to conflict (monthly, daily) [27], drought (Palmers Drought Severity Index, Standardised Precipitation Index, monthly) [28, 29], internally displaced persons (IDPs) (households, individuals, annual) [30], WASH (improved drinking water, piped water, improved sanitation, open defecation, basic hygiene, annual) [31], healthcare (total facilities, facilities per 100,000 people, annual) [27], population (total, annual) [32] and poverty (MPI, headcount ratio in poverty, intensity of deprivation among the poor, severe poverty and population vulnerable to poverty, annual) [27].

Here, several drought metrics were used, measured across multiple time windows. The benefits of using multiple metrics when investigating both drought and floods has been suggested in previous work [7]. The drought indices were used to measure relative dryness/wetness, not long-term drought changes, due to the short timescale of the cholera surveillance dataset. Using a drought metric, instead of raw precipitation or temperature data were selected to account for several environmental variables (temperature, precipitation and potential evapotranspiration) and to better present how the raw data translated into drier or wetter environments.

Covariate data were on a range of spatial and temporal scales, therefore administrative level one (state) was set as the spatial granularity (data on a finer spatial scale were attributed to administrative level 1) and the finest temporal scale possible for covariate selection, repeating values where needed for monthly and annual data (the temporal granularity of each dataset is shown above).

### Incidence and R

The 2018 and 2019 laboratory confirmed linelist data were used to calculate incidence. Incidence was calculated on a daily scale by taking the sum of the cases reported by state and date of onset of symptoms. This created a new dataset with a list of dates and corresponding daily incidence for each state. All analysis was completed in R with R Studio version 4.1.0. (packages "incidence" [33] & "EpiEstim" [34]).

Rather than using incidence as the outcome variable (which has less implicit assumptions), R was calculated, as it is more descriptive providing information on epidemic evolution (e.g., R = >1, cases are increasing), instead of new reported disease cases for a single time point. R was calculated from incidence using the parametric standard interval method, which uses the mean and the standard deviation of the standard interval (SI). SI is the time from illness onset in the primary case to onset in the secondary case and therefore impacts the evolution of the epidemic and speed of transmission. The SI for cholera is well-documented and there are several estimates in the literature [33–37]. To account for this reported variation in SI, a sensitivity analysis was conducted with SI set at 3, 5 and 8 days with a standard deviation of 8 days. The parametric method was used (vs the non-parametric which uses a discrete distribution), as the data can be adequately modelled by a normal probability distribution and has a fixed set of parameters.

Estimating R too early in an epidemic increases error, as R calculations are less accurate when there is lower incidence over a time window. A way to understand how much this impacts R values is to use the coefficient of variation (CV), which is a measure of how spread out the dataset values are relative to the mean. The lower the value, the lower the degree of variation in the data. A coefficient of variation threshold was set to 0.3 (or less) as standard, based on previous work [34]. To reach the CV threshold, calculation start date for each state was altered until the threshold CV was reached. States with <40 cases were removed, as states with fewer cases did not have high enough incidence across the time window to reach the CV threshold. Additionally, R values were calculated over monthly sliding windows, to ensure sufficient cases were available for analysis within the time window.

## Covariate selection and random forest models

Supervised machine learning algorithms such as decision-tree based algorithms, are now a widely used method for predicting disease outcomes and risk mapping [38, 39]. They work by choosing data points randomly from a training set and building a decision tree to predict the expected value given the attributes of these points. Transparency is increased by allowing the number of trees (estimators), number of features at each node split and resampling method to be specified. Random Forests (RF) then combines several decision trees into one model, which has been shown to increase predictive accuracy over single tree approaches, while also dealing well with interactions and non-linear relationships [40, 41].

The covariates listed above (conflict, drought, IDPs, WASH, healthcare, population and poverty) were first clustered to assist in the selection of covariates for model inclusion and to understand any multicollinearities. Despite RF automatically reducing correlation through subsetting data and tuning the number of trees and depth [39, 42], the process here lends support that the final model is measuring somewhat independent processes and not purely over-fitting the same patterns [38]. The clustering was based on the correction between the covariates meeting an absolute pairwise correlation of above 0.75. A secondary covariate selection process was run during preliminary analysis and acted as a method of validation. The process is detailed in S2 Text.

Random forest variable importance was used to rank all 22 clustered covariates. Variable importance provided an additional method of guiding the fitting of the best fit model, by testing the covariates which found the highest variable importance first. In this context, variable importance is a measure of the cumulative decreasing mean standard error each time a variable is used as a node split in a tree. The remaining error left in predictive accuracy after a node split is known as node impurity and a variable which reduces this impurity is considered more important.

Training (70% of data) and testing (30%) datasets were created to train the model and test the model's predictive performance. Random forest regression models (as opposed to classification models) were used since the outcome variable (R) is continuous. The parameters for training were set to repeated cross-validation for the resampling method, with ten resampling interactions and five complete sets of folds to complete. The model was tuned and estimated an optimal number of predictors at each split of 2, based on the lowest out-of-bag (OOB) error rate with RMSE used as the evaluation metric (package "caret" [43]).

A stepwise analysis was used to fit the models under each SI condition (3, 5 & 8 days), taking into consideration the covariate clustering and variable importance. One covariate was selected from each cluster, and all combinations of covariates were tested until the best-fit model was found. Models were assessed against each other in terms of predictive accuracy, based upon $R^2$ and RMSE. Predictions were then calculated on the testing dataset to compare incidence-based (R values calculated using the incidence data) vs covariate-based R values (R values calculated through model predictions). The terms, actual vs predicted was not used here, as all R values were modelled making the term "actual" misleading in this context. Model performance evaluations were built on multiple metrics including correlation, $R^2$ and RMSE.

Despite random forest models being accurate and powerful for predictions, they are easily over-fit (fitting to the testing dataset too closely or exactly) and therefore calculating error for the predictions are important. Little to no error in the predictions are an indication of over-fitting which can occur through predictions based off too small a dataset, more parameters than can be justified by the data and multicollinearity. Here, error was calculated using mean absolute error (MAE), where $y_i$ is the prediction and $x_i$ is the true value, with the total number of data points as n.

$$MAE = \frac{\sum_{i=1}^{n} |y_i - x_i|}{n}$$

## Nowcasting

The best fit model, in terms of predictive power according to the metrics above, was used to predict R for the remaining states which did not have sufficient reported cases to calculate R using incidence or had missing data for certain dates. Data for the best fit model covariates were collected for the states and missing dates from the sources given above. The data for the selected covariates are shown spatially in S1 Fig.

## Traffic-light system for cholera outbreak risk

The best fit model was then used to predict the traffic-light system for cholera outbreak risk, by manipulating the covariates values and using these to predict R. The traffic light system was defined by:

- Red—Covariate values which pushed R over 1

- Amber—Covariates values with predicted R around 1

- Green—Covariate values which predicted R below 1.

By using these three traffic-light scenarios, cholera outbreak triggers were identified based on the conditions of the four selected covariates. No specific R value had to be met for each traffic-light scenario, to account for the complexity of the relationships and non-linearity (S2 & S3 Figs). Due to there being no specific guidelines for each covariate in the scenario, the full range of values were presented, along with a median value, to increase the transparency of each scenario. To illustrate the historical trends between the best fit model covariates and the

R thresholds (R = >1, R <1), the data were split both spatially (by month) and temporally (by state) in S4 & S5 Figs.

## Spatial heterogeneities

To understand spatial differences in the relationship between the selected social and environmental extremes (conflict and PDSI) and cholera outbreak risk and the role pre-exiting vulnerabilities played in altering these relationships, six states were selected for additional analysis. These states were selected because they had either a clear positive or clear negative relationship with conflict or PDSI and R (PDSI is hypothesised to increase R at either end of the scale, +4/-4) and included Borno, Kaduna, Nasarawa, Ekiti, Lagos and Kwara (see S4 Fig). The processes above for predicting R under the three traffic-light scenarios was repeated for the six states but only PDSI and conflict values were manipulated, keeping the other three covariates at the mean value for R = >1 across the full dataset for the state. The spatial analyses identified the thresholds in conflict and PDSI needed to push R values below 1.

## Results

### Incidence and R

In Nigeria, there were 837 and 564 confirmed cholera cases for 2018 and 2019, respectively (out of 44,208 and 2,486 total cases for 2018 and 2019, respectively). The results from the sensitivity analysis including confirmed and suspected cases, proved model robustness and that the smaller dataset was not biasing the results. The geographic distribution of confirmed cases is shown in Fig 1 and are concentrated in the northeast of the country, with Adamawa, Borno, Katsina and Yobe having the highest burden. The number of cases declined steeply with age to a minimum in the 35–44 years category, before increasing again over 45 years. Whereas, cases were relatively evenly split by sex overall, with slightly more males affected in 2018 (51.6% male) and more females in 2019 (43.6% male) (Fig 2).

Six states for 2018 and two states for 2019 had sufficient cases to be included for R calculations, including Adamawa (2018 & 2019), Bauchi (2018), Borno (2018 & 2019), Gombe (2018), Katsina (2018) and Yobe (2018). Both the R values and the incidence data used to calculate R are shown temporally in Fig 3 for each state and year. Some states appear to have a peak in transmission around June-July, whereas others appear later during September to October.

### Covariate selection and random forest models

Twenty-one covariates were included in the clustering and variable importance analyses and were grouped into nine clusters. The clusters and variable importance (based on reducing node impurity) of each covariate are shown in Fig 4. Stepping through different covariate combinations, the best fit model included number of monthly conflict events, Multidimensional Poverty Index (MPI) (annual), Palmers Drought Severity Index (PDSI) (monthly) and improved access to sanitation (annual), fitted to R values with a serial interval of 5 days (standard deviation: 8 days). The fit of the incidence-based vs covariate-based R values (including error) are shown in Fig 5 and had a correlation of 0.87, with the model Root-Mean-Square Error (RMSE) at 0.33 and $R^2$ of 0.32.

### Nowcasting

Using the best fit model, R was predicted for the remaining 31 states which did not have sufficient cases to be included in the R calculations and any missing dates for the six states which were included. This created estimates of R for all 37 states on a monthly temporal scale for

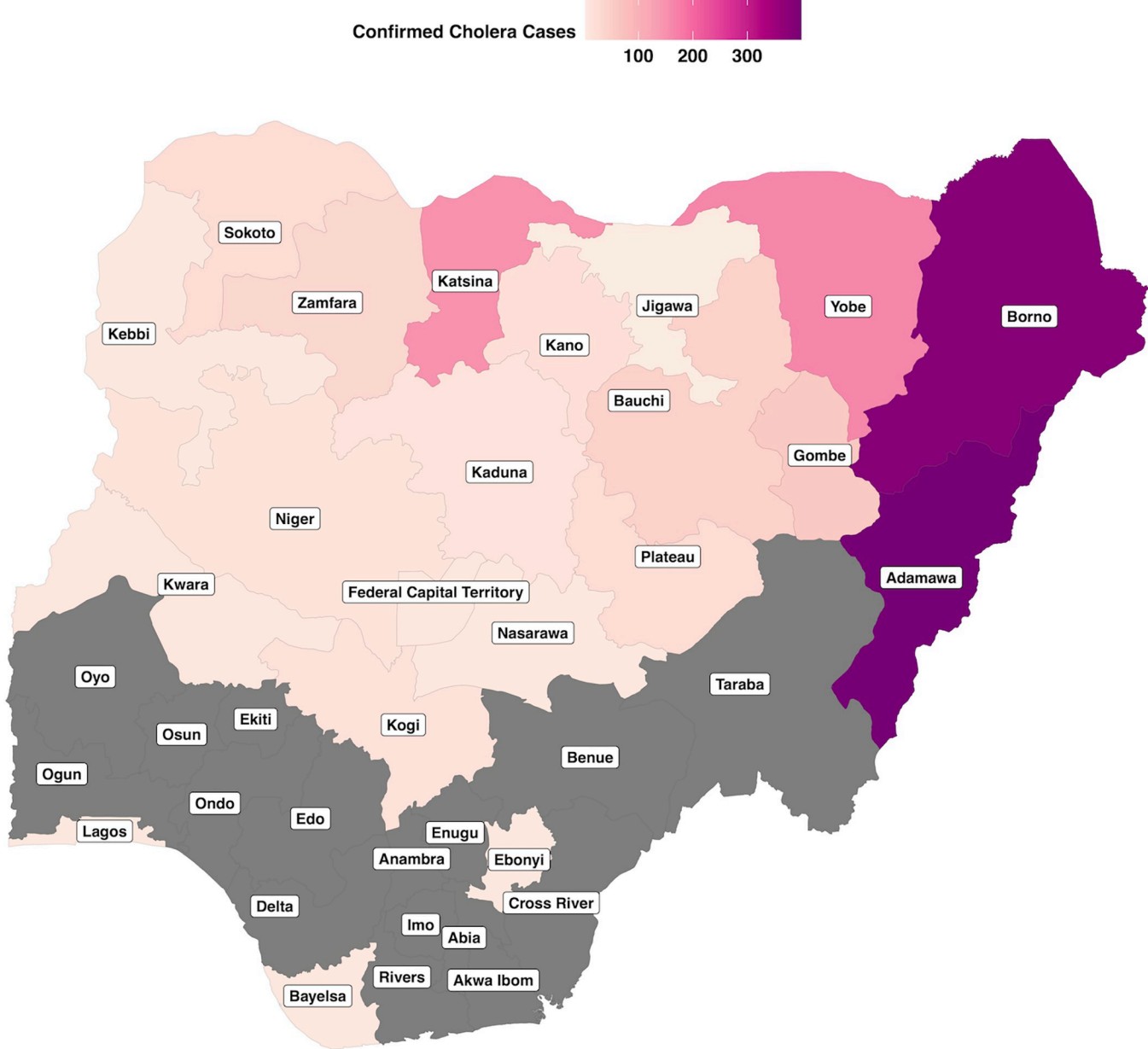

**Fig 1. Number of confirmed cholera cases by state for 2018 and 2019, grey indicates states that had no reported confirmed cases** [44].

2018 and 2019. The predictions provide further evidence that the model accurately predicts R, as the higher R values were in areas with known elevated cholera burden (northern and northeastern regions) and the states which only marginally fell below the threshold for R calculations (e.g., Niger, Sokoto and Taraba) (Fig 6).

## Traffic-light system for cholera outbreak risk

Fig 7 shows the predicted R values for the three traffic-light scenarios (Red = R over 1, Amber = R around 1 and Green = R less than 1) of cholera outbreak risk, based on the four selected covariates. Sanitation and MPI had a clear relationship with the R threshold, with consistently lower MPI (less poverty) and a higher proportion of people with access to sanitation

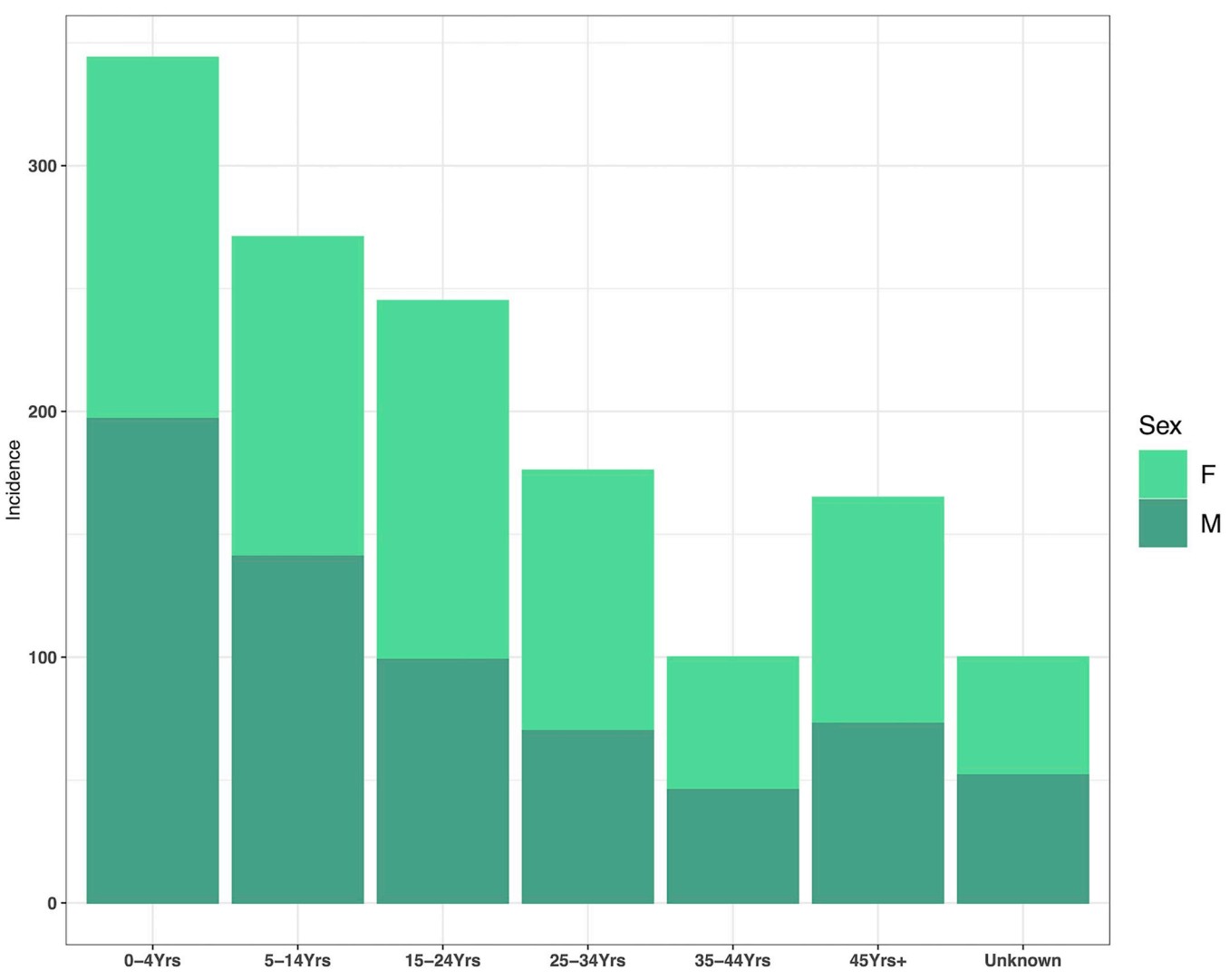

**Fig 2. Number of confirmed cholera cases by sex and age group for 2018 and 2019.**

seeing lower R values. R increased above 1 at 50% or lower for improved sanitation access and MPI values of above 0.32. The historical average sanitation level for R = >1 was 52.8% for the full dataset, whereas for R <1 it was 61.2%, for MPI the mean values were 0.27 and 0.13 for R = >1 and R <1, respectively.

In contrast, monthly conflict events and PDSI shows a less defined relationship, with conflict having a wide range of values in each of the three traffic-light scenarios. For PDSI and conflict, R values increased above 1 at around -1.1 for PDSI and monthly conflict events of 1.6. The historical spatial trends for conflict and PDSI are presented in S5 Fig and shows polarity in the relationships between the selected social and environmental extremes and R values, which differ between states.

## Spatial heterogeneities

**Conflict.** Borno and Kaduna were selected due to their clear positive relationship between conflict and R (increased conflict and R = >1). The three traffic-light scenarios created for

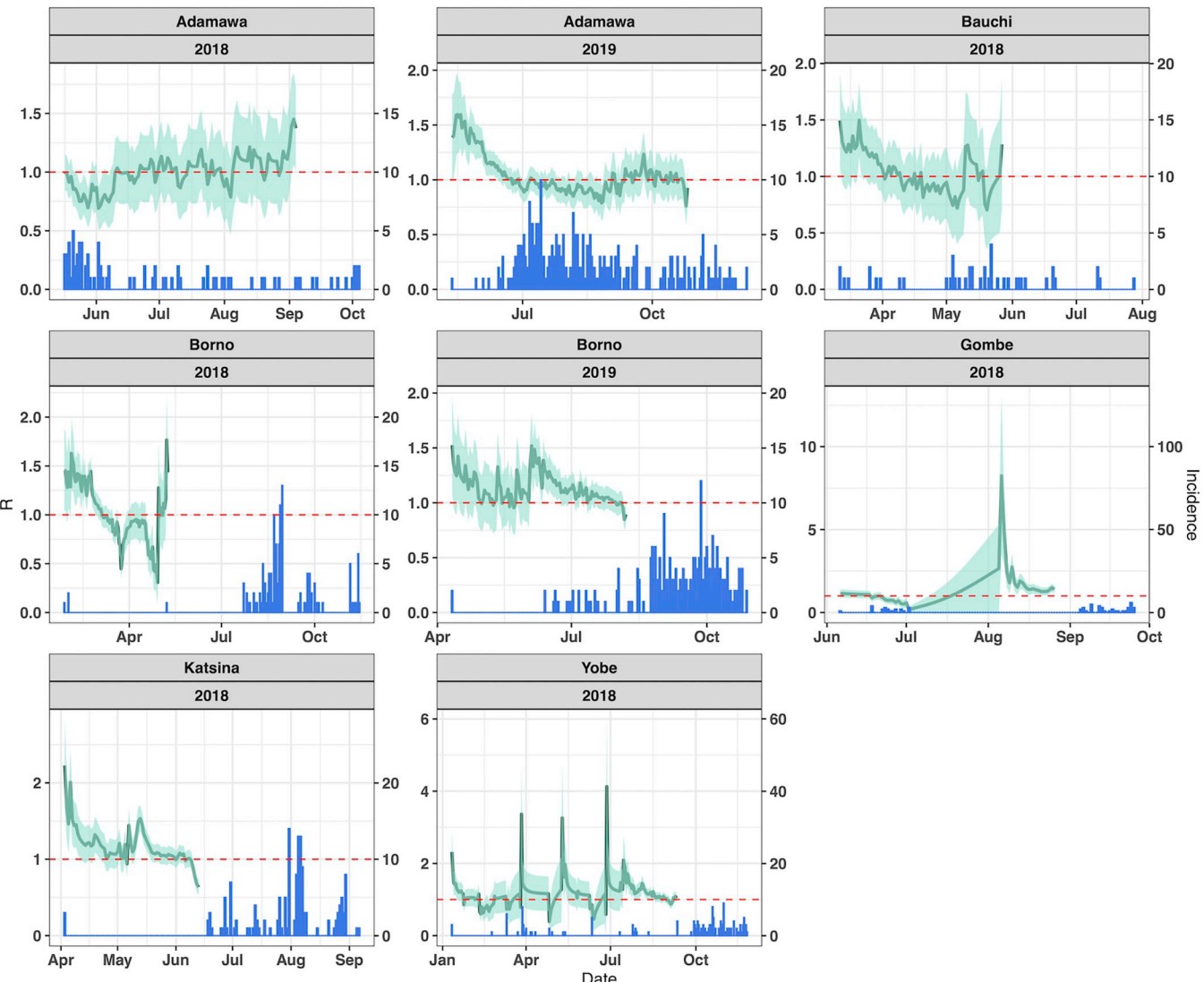

**Fig 3. R values over monthly sliding windows (line) calculated from the daily incidence (bar) of cholera.** The data used were only confirmed cholera cases for 2018 and 2019 of states which met the threshold equal to or more than 40 cases.

conflict in these two states found a consistently high cholera outbreak risk. The Green traffic-light scenario was relatively small, with only a narrow range of conflict values causing R values less than 1. Both Kaduna and Borno have high levels of poverty and low access to sanitation (40–41% access). For Borno, raising monthly conflict events from 1 to 2 increased R above 1, but an increase in access to sanitation from 41–46% pushed the R value back below one. This relationship continued in a stepwise pattern and in a similar way for MPI and drought but to a lesser degree. This showed that increasing sanitation and therefore decreasing vulnerability, allowed the states to adapt to increasing conflict and keep the R value below 1 (See S6 Fig).

**Drought.** Four states were investigated to evaluate the differences between extreme wetness (Lagos and Ekiti) and extreme dryness (Nasarawa and Kwara) and R values over 1. In contrast to Borno and Kaduna, all four states predicted consistently low R values (S7 & S8 Figs), a potential explanation for this is the high variable importance of PDSI (Fig 4) and the high levels of sanitation and low levels of poverty in all four states, contributing to overall

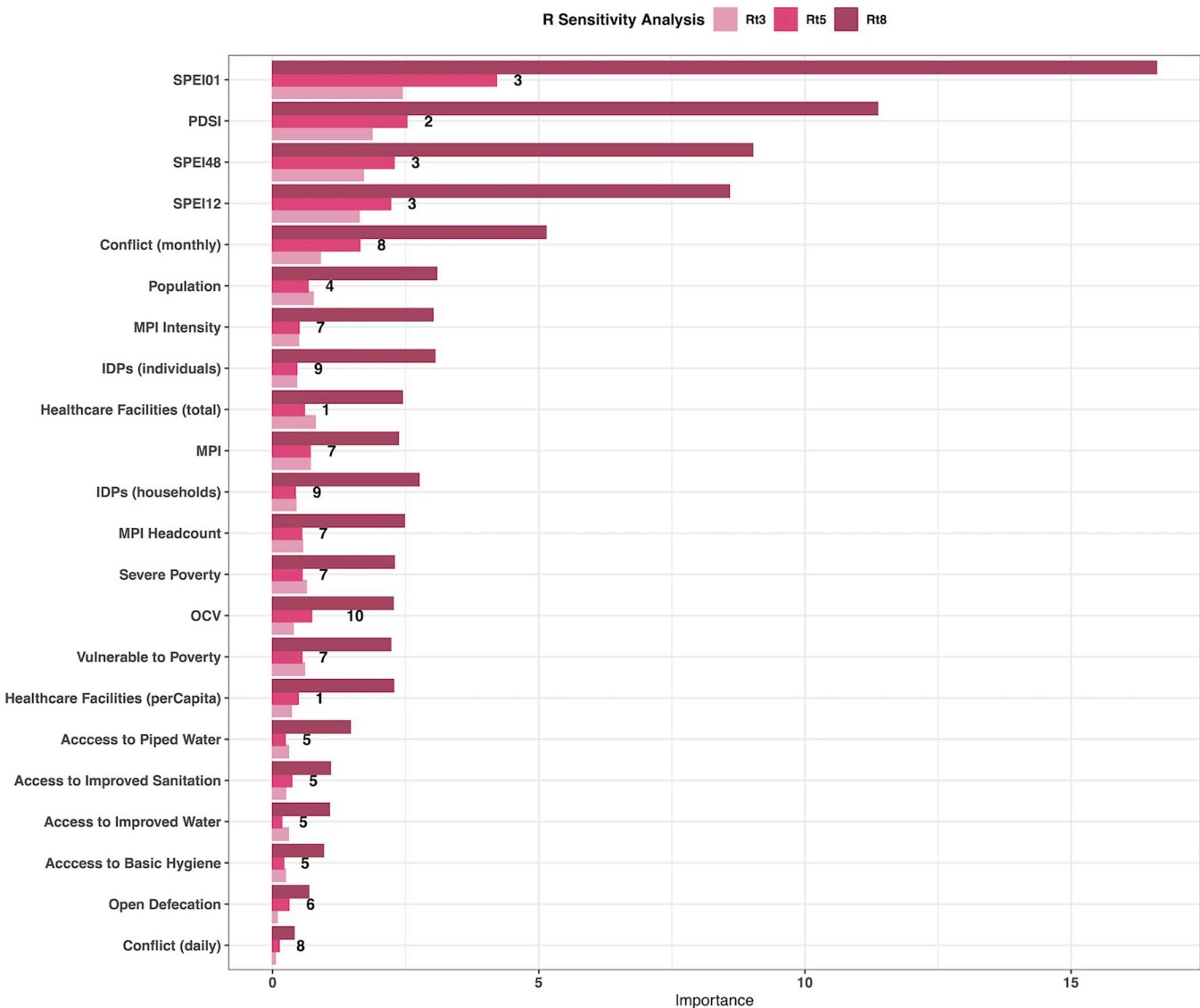

**Fig 4. The variable importance for the 21 tested for inclusion in the best fit model.** All three serial interval values tested are shown (Rt3–3 days, Rt5–5 days, Rt8–8 days) and the numbers represent the clusters. Variable importance is measured through node impurity (see Methods for details). SPEI01, 12, 48—Standardised Precipitation Index calculated on 1, 12 and 48 month scale. PDSI—Palmers Drought Severity Index. MPI—Multidimensional Poverty Index. IDP–Internally Displaced Persons. OCV–Oral Cholera Vaccination.

lower predicted levels of cholera. Therefore, the model was detecting a signal in only small changes in PDSI, that resulted in changing R values which have not been detected in other states with higher rates of poverty and lower levels of sanitation access. It also helps to highlight the multi-directionality of the relationship between PDSI and cholera transmission, with both extreme wetness and extreme dryness causing increases in R.

## Discussion

The results presented here show the importance of social and environmental extremes on cholera outbreaks in Nigeria, along with the importance of underlying vulnerability and socioeconomic factors. Of the 1,401 positive cases for Nigeria in 2018 and 2019, the northeast of the

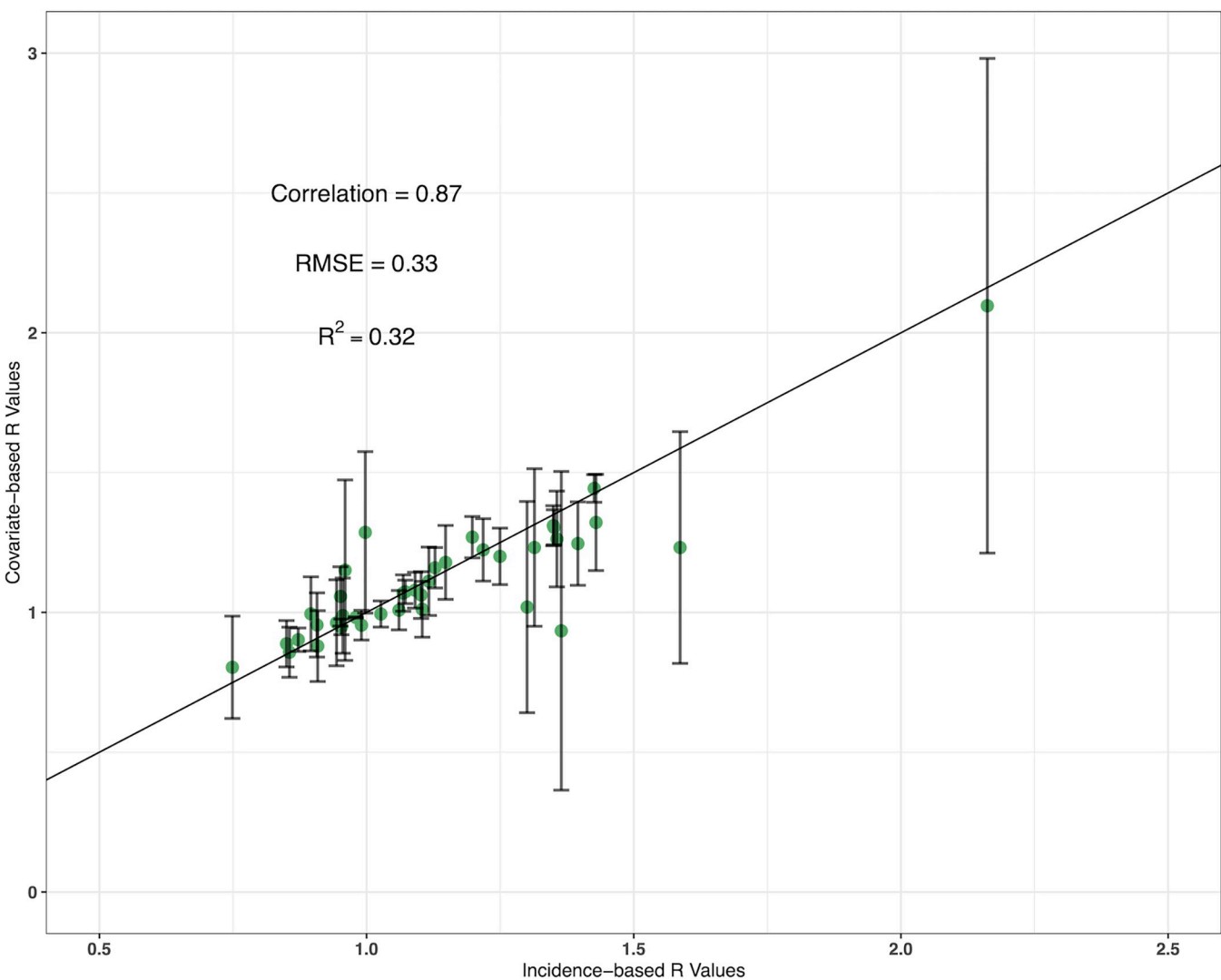

**Fig 5. Incidence-based vs covariate-based R values for the best fit model fitted to the testing dataset.** The error bars show mean absolute error and the line is a linear trend line intercepting at 0.

country and children under 5 carried the highest burden of disease, whereas there was minimal differentiation in cases between sex. Six states were used to calculate the R values, including Adamawa, Bauchi, Borno, Gombe, Katsina and Yobe. Twenty-one covariates were considered for model inclusion and the best fit model according to the selected model performance measures (variable importance based on node impurity, RMSE, $R^2$ and correlations) included monthly conflict events, percentage of the population with access to sanitation, MPI and PDSI. Using the best fit model, nowcasting was used to calculate the R values for the remaining thirty-one states which did not meet the threshold.

The predicted R values from the three traffic-light scenarios helped to shed light on the thresholds and triggers for raising R values above 1 in Nigeria. MPI and sanitation showed a well-defined relationship with R, with consistently higher access to sanitation and less poverty (lower MPI value) when R was less than 1. Thresholds which pushed R above one included decreasing access to sanitation below 50% and increasing the MPI above 0.32. Whereas the relationship between R and conflict events and PDSI appeared to vary spatially, with some

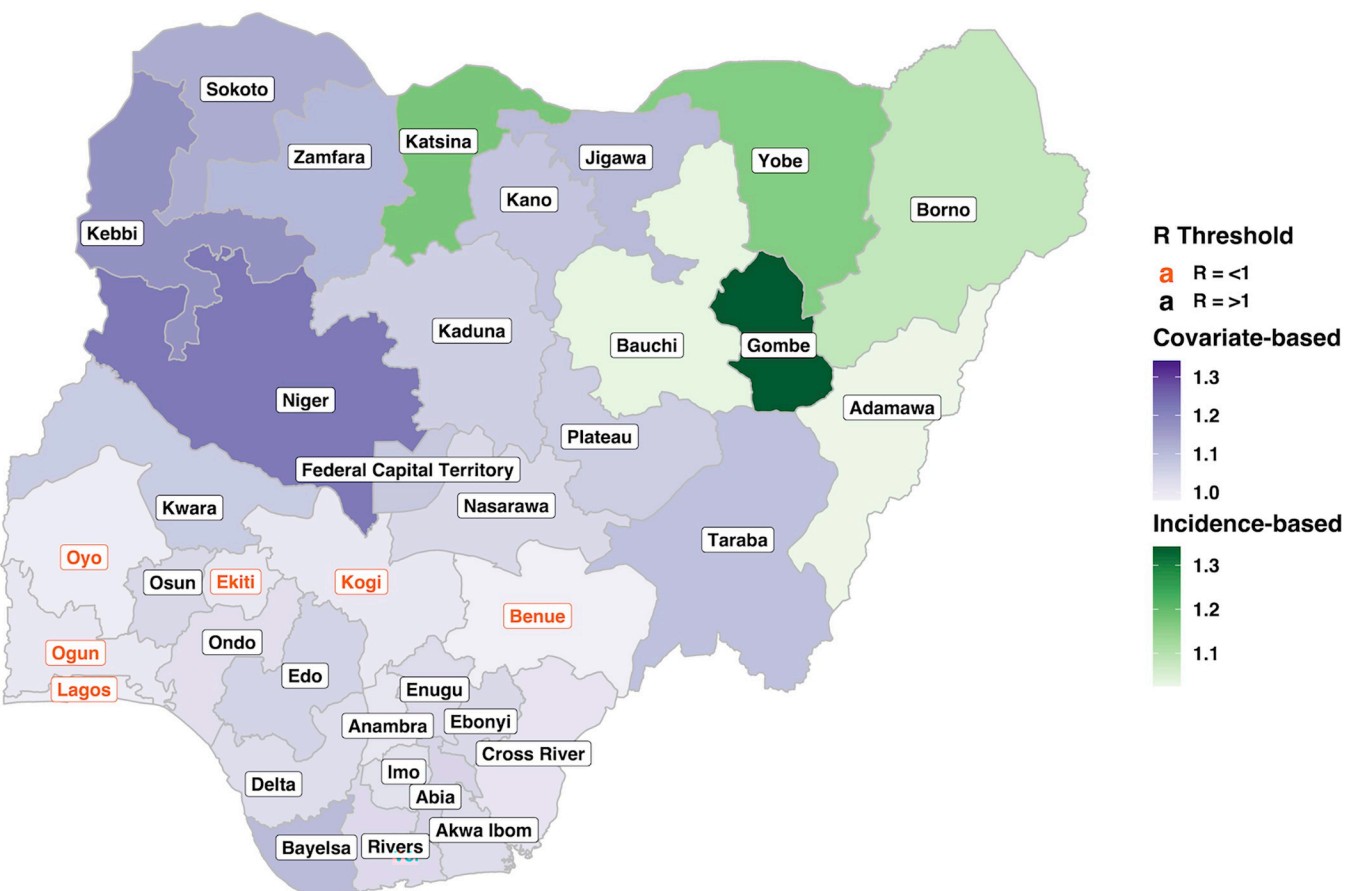

**Fig 6. Average R values for 2018 and 2019 for all 37 Nigerian states.** Incidence-based (green)—the six states which met the equal to or more than 40 case thresholds. Covariate-based (purple)—the 31 states which did not meet the threshold and had R predicted using the best fit model. State label colour shows which states had an average R of R = >1 (black) and R = <1 (orange) [44].

states showing a negative and some states a positive association. For these two covariates, the effect on R was largely dependent on the access to sanitation and poverty within the states, with high levels of sanitation and low poverty resulting in a decreased effect of PDSI and conflict. This showed that better sustainable development in the state acted as a buffer to social and environmental extremes and allowed people to adapt to these events better, due to less pre-existing vulnerability.

According to the World Bank [45], up to 47.3% (98 million people) of Nigeria's population live in multidimensional poverty. Poverty is a well-known risk factor for cholera, which is considered a disease of inequity [46], despite this, very few studies have suggested quantitative thresholds where poverty leads to disease. The results here suggest that states with an MPI value above 0.32 should be areas for poverty alleviation prioritisation (e.g., Kebbi, Sokoto, Yobe, Jigawa, Zamfara, Bauchi, Gombe, Katsina, Niger, Kano, Taraba, Borno and Adamawa). Poverty can result in several risk factor cascades, which puts people at risk of not just cholera but several other diseases. Examples of these risks include poor access to WASH [22, 23], inadequate housing [47], malnutrition [48] and overcrowding [47]. The expansion of sustainable development helps to reduce these risks and meeting or exceeding the Sustainable Development Goals would see significant gains in global health [49]. People living in poverty have fewer options and abilities to adapt to new and extreme situations, becoming trapped in the

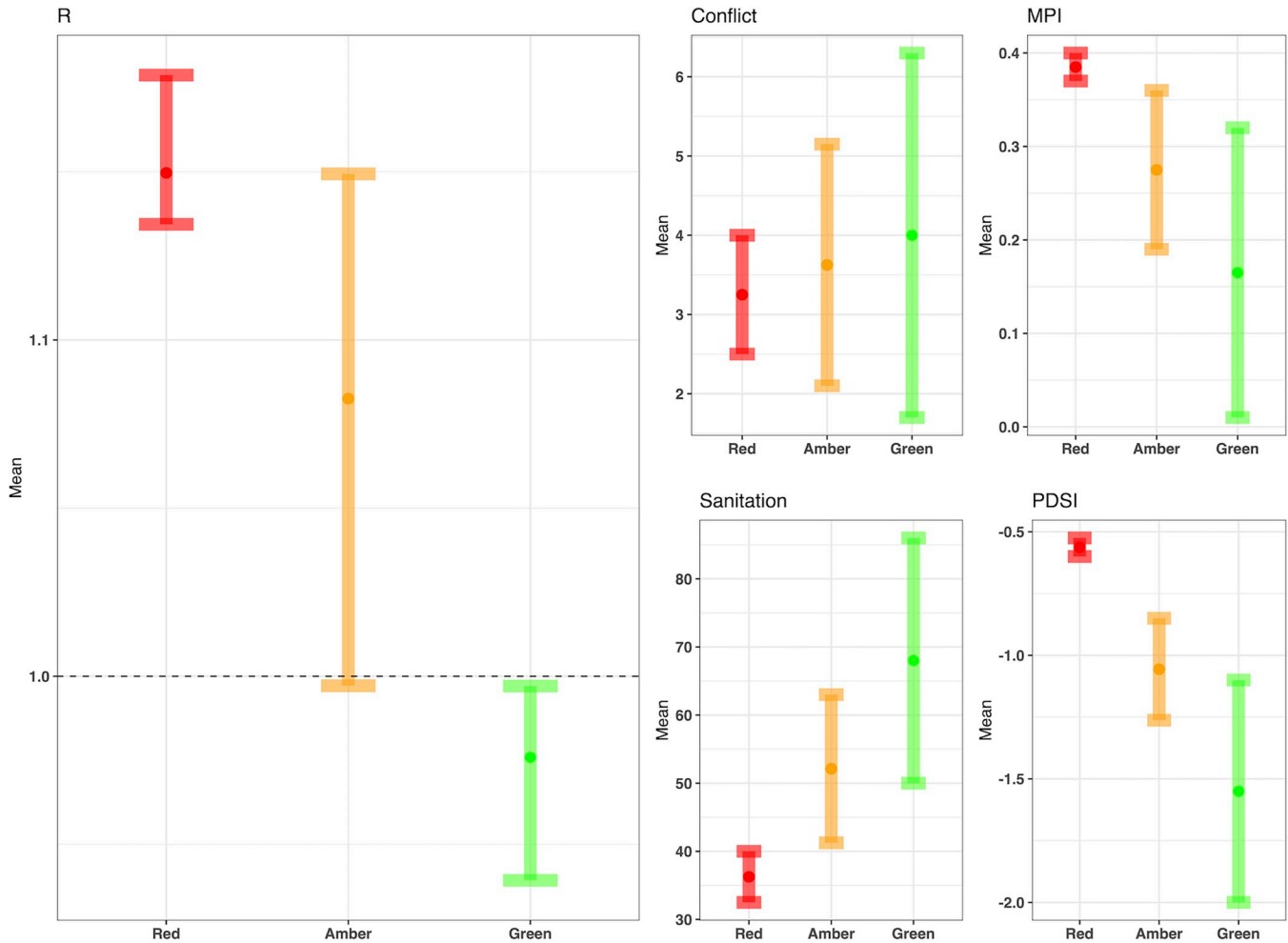

**Fig 7. Traffic-light system of cholera risk.** The three traffic-light scenarios (Red = R over 1, Amber = R around 1 and Green = R less than 1) for each of the four covariates in the best fit model and the corresponding predicated R value using the best fit model.

affected area or displaced to areas where their needs are not met. This provides further evidence for the need to reduce pre-existing vulnerabilities and to implement known techniques for reducing disasters [50, 51].

Poverty when measured in monetary terms alone can create issues due to its impact on the risk factors stated and is an advantage of using MPI as a poverty indicator. Nigeria's cash transfer scheme has allowed many Nigerians to meet the household income limit for poverty but there is a case for turning these funds and attention onto structural reform [52]. Nigeria's nationwide average access to sanitation is around 25%, therefore using these funds to increase access to sanitation may significantly improve health [53]. Currently, 73% of the enteric disease burden in Nigeria is associated with inadequate WASH [54] and here we show the need for expansion of sanitation to reduce cholera risks and the shocks of extremes on its transmission. The results here suggest that the expansion of sanitation would be particularly impactful for cholera control in states with <50% access, which currently includes all northern states. In a recent review on the implementation of non-pharmaceutical cholera interventions, there was generally a high acceptance of several WASH interventions. Despite this, education was key and building community relationships is needed to achieve this, such as understanding

cultural differences and barriers [55]. This is especially important in areas with conflict, where trust between the government and residents may have been lost [10].

Since 2002, Boko Haram (and Islamic State's West Africa Province) has been gaining a foothold and territory in northeastern Nigeria which has resulted in ongoing conflict, unrest and oppression of civilians [56]. Currently 5,860,200 people live in Borno state [57], where the fighting has been most concentrated. Millions of people comprise conflict-affected populations globally and there is an increasing proportion of people living in early post conflict areas [58]. This is significant in terms of health and disease, as conflict has known risk factors for cholera along with several other diseases [8, 10, 59] and can worsen several of the social risk factors discussed above. Here, conflict was included in the best fit model and in some states, highly influential in terms of cholera transmission. These results are the first to highlight the impacts of Boko Haram on a specific infectious disease, whereas previous research has focused more generally on public health [60–62]. The influence of conflict shows the need to incorporate and include conflict metrics in disease control research and policy in Nigeria and potentially in other conflict-affected countries. Providing services and protecting health in conflict zones is especially challenging and coordination across organisations in reporting and operations are needed to streamline resources and prevent duplication of services [63]. The traffic-light system used here helps highlight the need to protect basic services and reduce inequities in conflict situations to protect health and prevent outbreaks.

PDSI and several of the other drought indices tested here showed high variable importance but, in some states, had only marginal influence on R predictions when the PDSI values were manipulated. When analysing spatial differences between R and PDSI, the relationship appears to be multi-directional, with both extreme wetness (PDSI = +4) and extreme dryness (PDSI = -4) associated with R values above 1. Furthermore, access to sanitation and poverty were important in how PDSI impacted R, similar to the impacts of conflict. There is significant evidence to show that both droughts [7, 11] and floods [12, 64] can cause cholera outbreaks and elevated transmission and in Nigeria the risks of both the dry season and wet season have resulted in cholera outbreaks. Mechanisms through which this can occur includes a lack of water increasing risky drinking water behaviour and floods allowing for the dispersal of the pathogen [7, 65]. Despite this, drought is often a slow-onset disaster and PDSI is generally used to measure long-term change, therefore the limited timescale of the data used here means the results should be interpreted with caution. The insight presented shows that some states are impacted by either a relatively wetter or drier environment and suggests that in some states extra vigilance is needed. Continued work is essential to offset cholera risks related to droughts or floods through sanitation and hygiene, which can take significant time and resources [66].

Despite adapting the methodology to account for this, a potential limitation may be lagged effects of the covariates on cholera [67, 68]. Both long-term and short-term changes to the population may take time before changes in cholera transmission are evident. While some influences may be considered slow-onset or rapid-onset and therefore defining their beginning is subjective. Despite this, the incubation period of cholera is short (<2 hours—5 days) and previous research has suggested that acute impacts cause increases in cholera cases within the first week of the event [69–71]. Calculating R on monthly sliding windows and using monthly covariate data helped to reduce potential lagged effects on the R values, which would be captured if the one-week lag estimate is applicable here. Although beyond the scope of this research, the impacts of different lagged periods for several of these covariates and cholera outbreaks is an essential area of future research.

Cholera is considered an under-reported disease, and the lack of symptomatic cases means that many are likely to be missed. The data used here were also on a relatively short timescale and therefore is more accurate at presenting cholera at the current time in Nigeria, rather than

historically. Consequently, caution is needed when making generalisable conclusions. There are also incentives not to report cholera cases, due to travel restrictions and isolations and implications for trade and tourism [72]. During times of crisis, cholera may also be over-reported or more accurately represent the cholera burden in the area. This is due to the presence of cholera treatment centres, increased awareness among the population and healthcare workers and external assistance from non-governmental organisation, detecting cases that may have been missed previously [1, 8].

Despite the temporal (2 years) and spatial (6 states meeting the case threshold) limitations of the surveillance data, data of this detail is time consuming and difficult to collect in fragile settings and is the best data currently available to quantify cholera in Nigeria. Using confirmed cases only is necessary for modelling disease accurately, as in resource poor situations (outbreaks, conflicts) only a certain number of cases are confirmed, while it is very likely that several other intestinal pathogens could be causing disease. Therefore, the results and conclusions here are valid, novel and robust (presented through the absence of bias in the calculated error and S1 Text), if not more so, than models fit to longer but less accurate data sources [12, 20, 25]. Using accurate data were particularly important when fitting powerful predictive models, such as machine learning algorithms. The performance metrics such as the correlation between covariate and incidence-based R values, along with the predictions of R replicating the reality of cholera in Nigeria (e.g., southern states predicted lower R) suggest that the model accurately predicts cholera reproductive number across the country.

The Global Task Force on Cholera Control's 2030 target of reducing cholera deaths by 90% [73] will require acceleration of current efforts and significant commitment. Increasing cholera research and data are important in achieving this and the traffic-light system for cholera risk presented here sheds light on ways to reduce cholera outbreaks in fragile settings. The results here, although specific to a certain geographic area and timescale, highlight the importance of extreme events on cholera transmission and how reducing pre-existing vulnerability could offset the resultant cholera risk. Identifying specific targets and thresholds to avoid disease outbreaks enables targeted and therefore more successful policy strategies. This research is the first time several disaster types and measures of population vulnerability have been evaluated together quantitatively in terms of cholera and helps to further quantify the impacts of Boko Haram and conflict in Nigeria. We hope it shows the importance of doing so to gain a more accurate understanding of disease outbreaks in complex emergencies. Nigeria is currently working towards its ambitious goal of lifting 100 million people out of poverty by 2030 [52]. If it is successful, this could significantly improve health, increase quality of life and decrease the risks of social and environmental extremes.

## Supporting information

**S1 Text. Sensitivity analysis using confirmed and suspected cholera cases.** The analysis includes R calculations, variable importance and model fitting for the full dataset.
(DOC)

**S2 Text. Additional covariate selection using linear regression.**
(DOC)

**S1 Fig. Average values of the four covariates included in the best fit model.** By state, covariates included: **A**, monthly conflict events, **B**, Palmers Drought Severity Index (PDSI), **C**, percentage access to sanitation and **D**, Multidimensional Poverty Index (MPI) [69].
(TIFF)

**S2 Fig. Single predictor partial dependency plots for the covariates in the best fit model.**
Showing the relationships between **A**, monthly conflict events, **B**, access to sanitation, **C**, Palmers Drought Severity Index (PDSI) and **D**, Multidimensional poverty Index (MPI) and R.
(TIFF)

**S3 Fig. Multi predictor partial dependency plots for the covariates in the best fit model.**
Showing the relationships between **A**, Palmers Drought Severity Index (PDSI) & Multidimensional poverty Index (MPI), **B**, PDSI & Sanitation, **C**, Monthly conflict & MPI, **D**, Monthly conflict & Sanitation, **E**, Sanitation & MPI and R.
(TIFF)

**S4 Fig. Historical spatial trends between the selected social and environmental extremes (conflict and PDSI) and the R thresholds (R = >1, R <1).** The mean and standard error for the two covariates for the full dataset split by state and R threshold. The red "x" shows the states which were included in the sub-national analysis: Conflict (Borno and Kaduna), extreme wetness (Lagos and Ekiti), extreme dryness (Nasarawa and Kwara).
(TIFF)

**S5 Fig. Historical temporal trends between the best fit model covariates and the R thresholds (R = >1, R <1).** The mean and standard error for the four covariates included in the best fit model for the full dataset split by month and R threshold.
(TIFF)

**S6 Fig. Three traffic-light scenarios for conflict only and the corresponding predicted R values.** The other three (PDSI, Sanitation and MPI) covariate values were retained at the mean value for R = >1 for the full dataset (values shown in the plot) for **A**, Borno and **B**, Kaduna.
(TIFF)

**S7 Fig. Three traffic-light scenarios for PDSI (drier conditions) only and the corresponding predicted R values.** The other three (Conflict, Sanitation and MPI) covariate values were retained at the mean value for R = >1 for the full dataset (values shown in the plot) for **A**, Kwara and **B**, Nasarawa.
(TIFF)

**S8 Fig. Three traffic-light scenarios for PDSI (wetter conditions) only and the corresponding predicted R values.** The other three (Conflict, Sanitation and MPI) covariate values were retained at the mean value for R = >1 for the full dataset (values shown in the plot) for **A**, Ekiti and **B**, Lagos.
(TIFF)

## Acknowledgments

We would like to thank and acknowledgment the Nigeria Centre for Disease Control for providing the data used here and those who work for the NCDC who collected the data in the field. We would also like to thank Anwar Musah (University College London) and Kelly Elimian (Karolinska Institutet) for their guidance on cholera data for Nigeria and facilitating the partnership with NCDC. This work was supported by the Natural Environmental Research Council [NE/S007415/1], as part of the Grantham Institute for Climate Change and the Environment's (Imperial College London) Science and Solutions for a Changing Planet Doctoral Training Partnership. We also acknowledge joint Centre funding from the UK Medical Research Council and Department for International Development [MR/R0156600/1].

## Author Contributions

**Conceptualization:** Gina E. C. Charnley, Ilan Kelman, Katy A. M. Gaythorpe, Kris A. Murray.

**Data curation:** Sebastian Yennan, Chinwe Ochu.

**Formal analysis:** Gina E. C. Charnley.

**Funding acquisition:** Gina E. C. Charnley.

**Investigation:** Gina E. C. Charnley.

**Methodology:** Gina E. C. Charnley, Katy A. M. Gaythorpe, Kris A. Murray.

**Resources:** Sebastian Yennan.

**Supervision:** Ilan Kelman, Katy A. M. Gaythorpe, Kris A. Murray.

**Validation:** Sebastian Yennan, Chinwe Ochu, Ilan Kelman, Katy A. M. Gaythorpe, Kris A. Murray.

**Visualization:** Gina E. C. Charnley.

**Writing – original draft:** Gina E. C. Charnley, Sebastian Yennan, Chinwe Ochu, Ilan Kelman, Katy A. M. Gaythorpe, Kris A. Murray.

**Writing – review & editing:** Gina E. C. Charnley, Sebastian Yennan, Chinwe Ochu, Ilan Kelman, Katy A. M. Gaythorpe, Kris A. Murray.

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
