## [Decision Letter · Decision Letter 0]

23 Aug 2022

PGPH-D-22-00893

The impact of social and environmental extremes on cholera time varying reproduction number in Nigeria

Dear Dr. Charnley,

Thank you for submitting your manuscript to PLOS Global Public Health. After careful consideration, we feel that it has merit but does not fully meet PLOS Global Public Health’s publication criteria as it currently stands. Therefore, we invite you to submit a revised version of the manuscript that addresses the points raised during the review process.

We look forward to receiving your revised manuscript.

Kind regards,

Bernard Cazelles, Ph.D.

Academic Editor

Journal Requirements:

1. Please amend your online detailed Financial Disclosure statement. This is published with the article. It must therefore be completed in full sentences and contain the exact wording you wish to be published.

Please state the initials, alongside each funding source, of each author to receive each grant.

2. Please provide separate figure files in .tif or .eps format only and ensure that all files are under our size limit of 10MB.

4. All figures and supporting information files will be published under the Creative Commons Attribution License (creativecommons.org/licenses/by/4.0/). Authors retain ownership of the copyright for their article and are responsible for third-party content used in the article. 

Figure 1a, Figure 5, and S1 Figure: please (a) provide a direct link to the base layer of the map (i.e., the country or region border shape) and ensure this is also included in the figure legend; and (b) provide a link to the terms of use / license information for the base layer image or shapefile. We cannot publish proprietary or copyrighted maps (e.g. Google Maps, Mapquest) and the terms of use for your map base layer must be compatible with our CC-BY 4.0 license. 

Please upload any written confirmation as an 'Other' file type. It must clarify that the copyright holder understands and agrees to the terms of the CC BY 4.0 license; general permission forms that do not specify permission to publish under the CC BY 4.0 will not be accepted. Note that uploading an email confirmation is acceptable.

Additional Editor Comments (if provided):

The MS appears to be of some interest. Nevertheless, certain points should be better presented, such as the role of environmental factors and the mechanisms by which they act. The influence of the relatively short length of the data used should also be better discussed

It would also be important to place the results presented in the context of current knowledge about cholera epidemics by emphasizing more clearly the novelty of the results presented.

Reviewers' comments:

Reviewer's Responses to Questions

**Comments to the Author**

1. Does this manuscript meet PLOS Global Public Health’s publication criteria? Is the manuscript technically sound, and do the data support the conclusions? The manuscript must describe methodologically and ethically rigorous research with conclusions that are appropriately drawn based on the data presented.

Reviewer #1: Yes

Reviewer #2: Yes

2. Has the statistical analysis been performed appropriately and rigorously?

Reviewer #1: Yes

Reviewer #2: No

3. Have the authors made all data underlying the findings in their manuscript fully available (please refer to the Data Availability Statement at the start of the manuscript PDF file)?

Reviewer #1: Yes

Reviewer #2: Yes

4. Is the manuscript presented in an intelligible fashion and written in standard English?

Reviewer #1: Yes

Reviewer #2: Yes

5. Review Comments to the Author

Reviewer #1: Charnley et al. Soubmitted a manuscript entitled “The impact of social and environmental extremes on cholera time varying reproduction number in Nigeria” in PLOS Global Public Health.

The authors in this study aimed to understand the risks and thresholds for cholera outbreaks and extreme events in Nigeria, taking into consideration pre-existing vulnerabilities.

They estimated time varying reproductive number (R) from cholera incidence in Nigeria and used a machine learning approach to evaluate its association with extreme events (conflict, flood, drought) and pre-existing vulnerabilities (poverty, sanitation, healthcare). Then, they created a traffic-light system for cholera outbreak risk, using three hypothetical traffic-light scenarios (Red, Amber and Green) and used this to predict R.

They found that reducing poverty and increasing access to sanitation lessened vulnerability to increased cholera risk caused by extreme events (monthly conflicts and the Palmers Drought Severity Index).

My overall impression after the reading of the manuscript and the SI is good. The ms is well written, structured and results clearly illustrated.

My overall concerns are about the contribution of this research to improve the knowledge of cholera.

The conclusions drawn by the authors are already well known and described in the literature about the reducing effect of increase sanitation and decrease poverty on the cholera transmission as well as the effect of extreme climatic events as floods or droughts, or conflicts.

Furthermore, this study does not consider prevention interventions or vaccination campaigns.

The authors used an innovative statistical approach of machine learning in this study, even exploring the spatial heterogeneities. But the study limitations negatively impact the value of the results (2 years dataset, low number of cholera cases).

Despite those limitations, the authors have taken many precautions to avoid bias in the methodology, the results and in the conclusions.

Unfortunately, a small credit can be given to the fact that this research is the first time several disaster types and measures of population vulnerability have been evaluated together quantitatively in terms of cholera.

I don’t agree with the authors hope this study shows the importance of doing so to gain a more accurate understanding of disease outbreaks in complex emergencies.

What this study could help is to draw a picture at a given time for in the specific context of cholera in Nigeria at that time by a quantitative approach. It can serve to highlight specific vulnerabilities to elaborate specific recommendations to the public health authorities and political decision makers for the disease prevention.

This is why I consider the paper for publication in this journal.

But in the context of an emergencies, when some health authorities already struggle with the case detection, they put human and financial efforts on outbreak responses in the field instead of complex modeling.

Reviewer #2: The Reproductive number approach to categorize and classify situations leading to cholera outbreaks and various risk factors is a novel attempt. The application of machine learning in this approach is also at the cutting edge of research. I commend the authors for taking on this effort.

However, there are several substantial weaknesses that this manuscript suffer from,

which need to be addressed before the manuscript is strong enough for publication.

1. Although the study attempts to related both environmental and socio-economic extremes to cholera outbreaks, in reality, the paper mostly focuses on the social, political, and economic forcing and barely on environmental extremes.

2. The environmental variables are not properly developed. PDSI is a variable that is most appropriate to reflect long term change – and not really a daily process - only two years of daily data seems insufficient for understanding the trajectory of extreme droughts with daily scale cholera risk / reproductive ratio.

3. In addition, the mechanisms or pathways of how the environmental variables might play a role in cholera transmission or reproductive ratios – is virtually missing. I assume there is an underlying association with water scarcity and cholera that is implied here. But the reproductive number is a variable that can be affected in many facets of water scarcity – none of which have been envisioned, explained or discussed in this manuscript.

4.The reproductive number itself is a simulated variable here, although simulated in two different ways. But there is no way to validate the patterns without looking at pathways. Especially, the simulations over the western states with such little data, except Borno and Adamawa, are very hard to justify.

5. The above-mentioned issue with calculating reproductive ratios are reflected through the limited correlation seen between covariate and incidence-based R calculations. The threshold values around R = 1 are also not clear. There should be clear guidelines what those thresholds are for the dry or the saline conditions.

6. PLOS authors have the option to publish the peer review history of their article (what does this mean?). If published, this will include your full peer review and any attached files.

**Do you want your identity to be public for this peer review?** For information about this choice, including consent withdrawal, please see our Privacy Policy.

Reviewer #1: **Yes: **Guillaume CONSTANTIN DE MAGNY

Reviewer #2: No

---

## [Decision Letter · Decision Letter 1]

14 Oct 2022

PGPH-D-22-00893R1

The impact of social and environmental extremes on cholera time varying reproduction number in Nigeria

Dear Dr. Charnley,

Thank you for submitting your manuscript to PLOS Global Public Health. After careful consideration, we feel that it has merit but does not fully meet PLOS Global Public Health’s publication criteria as it currently stands. Therefore, we invite you to submit a revised version of the manuscript that addresses the points raised during the review process.

We look forward to receiving your revised manuscript.

Kind regards,

Bernard Cazelles, Ph.D.

Academic Editor

Journal Requirements:

1. Please insert an Ethics Statement at the beginning of your Methods section, under a subheading 'Ethics Statement'. It must include:

i) The approval number(s), or a statement that approval was granted by the named board(s) 

ii) (for human participants/donors) - A statement that formal consent was obtained (must state whether verbal/written) OR the reason consent was not obtained (e.g. anonymity). NOTE: If child participants, the statement must declare that formal consent was obtained from the parent/guardian.

Additional Editor Comments (if provided):

The reviewer 2 has raised some important gaps in our manuscript that it is important that you address...

Reviewers' comments:

Reviewer's Responses to Questions

**Comments to the Author**

1. If the authors have adequately addressed your comments raised in a previous round of review and you feel that this manuscript is now acceptable for publication, you may indicate that here to bypass the “Comments to the Author” section, enter your conflict of interest statement in the “Confidential to Editor” section, and submit your "Accept" recommendation.

Reviewer #1: All comments have been addressed

Reviewer #2: All comments have been addressed

2. Does this manuscript meet PLOS Global Public Health’s publication criteria? Is the manuscript technically sound, and do the data support the conclusions? The manuscript must describe methodologically and ethically rigorous research with conclusions that are appropriately drawn based on the data presented.

Reviewer #1: Yes

Reviewer #2: Yes

3. Has the statistical analysis been performed appropriately and rigorously?

Reviewer #1: Yes

Reviewer #2: Yes

4. Have the authors made all data underlying the findings in their manuscript fully available (please refer to the Data Availability Statement at the start of the manuscript PDF file)?

Reviewer #1: Yes

Reviewer #2: Yes

5. Is the manuscript presented in an intelligible fashion and written in standard English?

Reviewer #1: Yes

Reviewer #2: Yes

6. Review Comments to the Author

Reviewer #1: Thanks to the authors to have addressed all the comments and improved significantly the MS as well with all the modifications.

Reviewer #2: The authors have not advanced their case with respect to the variable selections, the model results, and the claimed conclusions. Machine Learning (ML) ‘can’ be a powerful tool – if it has enough good data to train the model – which is not the case here. Just using Machine learning cannot be the end itself. Thus, while focusing on a smaller high quality confirmed dataset has its merits – it also lends to the conclusion that ML is a good choice here.

The ML will fit to whatever you give it to fit. If you fit a smaller dataset, it will bias itself to that specific scenario. Using that ‘best fit’ model to predict the R values of other reasons seems a very erroneous approach to begin with. And there are little data to validate these other regions. High correlation can also result from a smaller number of data points, we all understand that.

Even if we are confident Random Forest is the way to go … the ranking of the variables in both the primary and the secondary selection process lead to the environmental variables (PDSI and other SPEI indices). The MPI, socio-economic, or sanitation related variables are all way down the list. While the model has its findings, the authors have not interpreted or conveyed it correctly. Sanitation coverage is universally low in Nigeria. How the role of the droughts make the pathogen transmit is the real story here!

The concluding sentences in the abstract “We found that reducing poverty and increasing access to sanitation lessened vulnerability to increased cholera risk caused by extreme events (monthly conflicts and the Palmers Drought Severity Index” and then “The results here therefore add further evidence of the need for sustainable development for disaster prevention and mitigation and to improve health and quality of life.” Thus appear rather generic and not supported by the evidence.

Please highlight the main findings of the models as the data suggest and not the way they help fit a narrative. Similarly, the added sentence in the discussion “The results here suggest that the expansion of sanitation would be particularly impactful or cholera control for those in states with <50% access, which currently includes all northern states.” Is almost universal. Please bring forward the discussion on the droughts (and water scarcity) play a central role in cholera progression in Nigeria … where safe water and sanitation is universally low to begin with. Improving local water and sanitation access will help only if Nigeria can handle the extreme events such as droughts well …

7. PLOS authors have the option to publish the peer review history of their article (what does this mean?). If published, this will include your full peer review and any attached files.

**Do you want your identity to be public for this peer review?** For information about this choice, including consent withdrawal, please see our Privacy Policy.

Reviewer #1: **Yes: **Guillaume CONSTANTIN DE MAGNY

Reviewer #2: No

---

## [Decision Letter · Decision Letter 2]

11 Nov 2022

The impact of social and environmental extremes on cholera time varying reproduction number in Nigeria

PGPH-D-22-00893R2

Dear Ms Charnley,

We are pleased to inform you that your manuscript 'The impact of social and environmental extremes on cholera time varying reproduction number in Nigeria' has been provisionally accepted for publication in PLOS Global Public Health.

Best regards,

Bernard Cazelles, Ph.D.

Academic Editor

Reviewer's Responses to Questions

**Comments to the Author**

1. If the authors have adequately addressed your comments raised in a previous round of review and you feel that this manuscript is now acceptable for publication, you may indicate that here to bypass the “Comments to the Author” section, enter your conflict of interest statement in the “Confidential to Editor” section, and submit your "Accept" recommendation.

Reviewer #2: All comments have been addressed

2. Does this manuscript meet PLOS Global Public Health’s publication criteria? Is the manuscript technically sound, and do the data support the conclusions? The manuscript must describe methodologically and ethically rigorous research with conclusions that are appropriately drawn based on the data presented.

Reviewer #2: Yes

3. Has the statistical analysis been performed appropriately and rigorously?

Reviewer #2: Yes

4. Have the authors made all data underlying the findings in their manuscript fully available (please refer to the Data Availability Statement at the start of the manuscript PDF file)?

Reviewer #2: Yes

5. Is the manuscript presented in an intelligible fashion and written in standard English?

Reviewer #2: Yes

6. Review Comments to the Author

Reviewer #2: The reviewers have made necessary changes and improvement to the manuscript.

7. PLOS authors have the option to publish the peer review history of their article (what does this mean?). If published, this will include your full peer review and any attached files.

**Do you want your identity to be public for this peer review?** For information about this choice, including consent withdrawal, please see our Privacy Policy.

Reviewer #2: No
